# RoMA: a Method for Neural Network Robustness Measurement and Assessment Conference Submission

## Abstract

Neural network models have become the leading solution for a large variety of tasks, such as classification, language processing, protein folding, and others. However, their reliability is heavily plagued by *adversarial inputs*: small input perturbations that cause the model to produce erroneous outputs. Adversarial inputs can occur naturally when the system's environment behaves randomly, even in the absence of a malicious adversary, and are a severe cause for concern when attempting to deploy neural networks within critical systems. In this paper, we present a new statistical method, called *Robustness Measurement and Assessment (RoMA)*, which can measure the expected robustness of a neural network model. Specifically, RoMA determines the probability that a random input perturbation might cause misclassification. The method allows us to provide formal guarantees regarding the expected frequency of errors that a trained model will encounter after deployment. Our approach can be applied to large-scale, black-box neural networks, which is a significant advantage compared to recently proposed verification methods. We apply our approach in two ways: comparing the robustness of different models, and measuring how a model's robustness is affected by the magnitude of input perturbation. One interesting insight obtained through this work is that, in a classification network, different output labels can exhibit very different robustness levels. We term this phenomenon *categorial robustness*. Our ability to perform risk and robustness assessments on a categorial basis opens the door to risk mitigation, which may prove to be a significant step towards neural network certification in safety-critical applications.

## 1 INTRODUCTION

In the passing decade, deep neural networks (DNNs) have emerged as one of the most exciting and innovative developments in computer science, allowing computers to outperform humans in various classification tasks. However, a major disadvantage of the DNN approach is the existence of *adversarial inputs* Goodfellow et al. (2014): inputs which are very close (according to some metrics) to a correctly-classified input, but which are misclassified themselves. It has been observed that most state-of-the-art DNNs are highly vulnerable to adversarial inputs Carlini & Wagner (2017); and it has been suggested that adversarial inputs are an inescapable part of the neural network architecture, and are thus not an issue that can be solved entirely Ilyas et al. (2019). In this perspective, it is crucial to find a method for containing this phenomenon and mitigating the risk that it causes; especially in order to allow DNNs to be deployed in safety-critical settings (e.g., automotive, aerospace, trains, and medical devices), where regulatory requirements and public opinion set a high bar for reliability.

As the impact of the AI revolution is becoming evident, regulatory agencies are starting to address the challenge of safely integrating DNNs into safety-critical systems — by forming workgroups to create the needed guidelines. Notable examples in the European Union include SAE G-34 and EUROCAE WG-114 Pereira & Thomas (2020); Vidot et al. (2021); and the efforts made by the European Union Safety Agency (EASA), which is responsible for civil aviation safety, and which has published a road map for certifying AI-based systems European Union Aviation Safety Agency (2020). These efforts highlight the dire need for certification methods for DNN-based systems.

The existence of adversarial inputs does not seem to be a deal-breaker as far as regulatory agencies are concerned; indeed, these agencies are used to certifying systems with components that might fail due to an unexpected hazard. A common example is the certification of jet engines for civilian aircraft, with a known *mean time between failures* (*MTBF*). Based on the MTBF value, there exist certification processes that perform *functional hazard analysis* (FHA) and risk mitigation FAA (2021). To perform similar processes for DNN-based systems, one needs to assess the likelihood of failure (i.e., an adversarial input), but this essential ability is still missing.

In this paper, we attempt to address this crucial gap by introducing a novel, straightforward, and scalable statistical method for measuring the probability that a DNN classifier will misclassify inputs. The method, which we term *Robustness Measurement and Assessment* (*RoMA*), assumes that the misclassification of the neural network is not due to a malicious attack, but rather due to random perturbations of the input, which occur naturally as part of the system's operation. Under this assumption, RoMA can be used to measure the model's robustness to randomly-produced adversarial inputs. The proposed method is useful for several applications, such as comparing the robustness of multiple models and picking the best one, or for checking the impact of various configurable parameters (e.g., the number of training epochs, or the magnitude of the input perturbation) on the resulting model's robustness.

RoMA is a method for estimating rare events in a large population — in our case, adversarial inputs within a space of inputs that are generally correctly classified. The method relies on the properties of *normal distributions*. If it is known that the rare events (adversarial inputs) are distributed normally, it is usually sufficient to sample a few thousands random points, use them to draw a Gaussian curve, and then use the normal distribution function to evaluate the probability of a rare event in the population. Unfortunately, adversarial inputs are often not distributed normally in the population; and to overcome this difficulty, RoMA first applies a statistical transformation, after which the distribution often becomes normal and can be analyzed.

At a high level, RoMA consists of the following steps:

1. for an arbitrary input point $x_0$, we randomly sample $n$ perturbations of $x_0$ (usually, a few thousands), obtaining a set of perturbed inputs $\{x_0^1, \ldots, x_0^n\}$;

2. we evaluate the DNN on each $x_0^i$, obtaining the corresponding outputs $\{y_0^1, \ldots, y_0^n\}$;

3. for each $y_0^i$ that is classified incorrectly (that is, which indicates a labeling different than that of $x_0$), we collect the maximal entry $c^i$ of $y_0^i$. This value is the confidence score assigned by the DNN to the incorrect label;

4. if needed, we apply a statistical transformation called Box-Cox Box & Cox (1982) to normalize the distribution of $c^i$ values; and

5. if the distribution is now normal, we use the properties of the normal distribution function to calculate the probability for an adversarial input around $x_0$.

A key component in our method is the Box-Cox statistical power transformation, which is a well-established method, and which does not pose any restrictions on the DNN in question (e.g., Lipschitz continuity, certain kinds of activation functions, or a specific network topology). Further, the method does not require access to the network's topology or weights, and is thus applicable to black-box DNNs.

We implemented our method as a proof-of-concept tool, and evaluated it on standard DNN architectures: VGG16 Simonyan & Zisserman (2015), Resnet He et al. (2016), and Densenet Huang et al. (2017a), all trained on the CIFAR10 data set Krizhevsky et al. (2009). We used RoMA to compare the robustness of these DNN models, and found, as expected, that a higher number of epochs leads to a higher robustness score. Additionally, we used RoMA to measure how the allowed magnitude of perturbation affects the robustness of a DNN model, and computed the rate at which robustness deteriorates as the perturbation level increases. Finally, using RoMA, we found that the categorial robustness score of a DNN (i.e., the robustness score of inputs labeled as a particular category) *varies significantly* among the different categories. This finding could allow users and regulators to specify different acceptable robustness thresholds for each target category, instead of a single global threshold, which may be more difficult to obtain.

To summarize, our main contributions are:

- Introducing RoMA: a new method for measuring the robustness of a DNN model. The new method is scalable and can run on black-box DNNs.

- Comparing the robustness of multiple state-of-the-art DNN models.

- Using RoMA to measure the effect of perturbation level has on the robustness of the DNN model.

- Introducing the notion of *categorial robustness*, which is a measure of robustness computed for each target label.

**Related work.** The topic of adversarial robustness has been studied extensively. Some notable approaches for estimating a model's robustness include:

- Statistical approaches that evaluate the probability of encountering an adversarial input in the population. In a recent paper Huang et al. (2021), Huang et al. use random sampling, which is similar in spirit to RoMA, but which assumes that the DNN's adversarial inputs are distributed normally. As we demonstrate later, this is often not the case. In another paper, Webb at al. Webb et al. (2018) use a sampling method called *multi-level splitting*, which provides no formal guarantee of the DNN's robustness. Mangal et al. Mangal et al. (2019) use *importance sampling*, which might be biased due to lack of sampling in areas of the population that are deemed *unimportant*. Moreover, this approach assumes that the network's output is Lipschitz-continuous, which limits its applicability. In contrast, RoMA requires no Lipschitz-continuity assumptions, does not assume a-priori that the adversarial inputs are distributed normally, and provides rigorous robustness guarantees.

- Formal-verification based approaches Katz et al. (2017); Wang et al. (2018); Jacoby et al. (2020), which allow for computing a DNN's exact adversarial robustness score. These approaches typically convert the problem into a constraint satisfiability problem, and then apply search and deduction procedures to solve it efficiently. However, verification-based approaches afford only limited scalability, and operate strictly on white-box DNNs. In contrast, RoMA is a scalable technique, and can operate on black-box DNNs.

- Approaches for computing an estimate bound on the probability that a classifier's margin function exceeds a given value Weng et al. (2019); Anderson & Sojoudi (2020); Dvijotham et al. (2018). These analyses focus on the worst-case behavior, thus producing bounds that might be inadequate for regulatory certification. In contrast, RoMA focuses on the average case, which is more realistic in many application domains.

**Outline.** We begin with some needed background on adversarial robustness in Section 2. We then describe our proposed method for measuring adversarial robustness in Section 3, followed by a description of our evaluation setup in Section 4. In Section 5 we summarize and discuss our results.

## 2 BACKGROUND

**Neural Network.** A neural network $N$ is a function $N : \mathbb{R}^n \to \mathbb{R}^m$, which maps a real-valued input vector $\boldsymbol{x} \in \mathbb{R}^n$ to a real-valued output vector $\boldsymbol{y} \in \mathbb{R}^m$. For classification networks, which is our subject matter here, the entries of $\boldsymbol{y}$ represent confidence scores for $m$ possible classes; and $\boldsymbol{x}$ is classified as label $l$ if $\boldsymbol{y}$'s $l$'th entry has the highest score; i.e., if $\arg\max(N(\boldsymbol{x})) = l$. We use $c(\boldsymbol{x})$ to denote this highest score, i.e. $c(\boldsymbol{x}) = \max(N(\boldsymbol{x}))$.

**Local Adversarial Robustness.** The local adversarial robustness of a DNN is a measure of how resilient that network is against adversarial perturbations to specific inputs. Intuitively, a network with high robustness behaves "smoothly", i.e., small perturbations to its input do not cause significant spikes in its output. More formally Bastani et al. (2016); Huang et al. (2017b):

**Definition 1.** *A DNN $N$ is $\epsilon$-locally-robust at input point $\boldsymbol{x}_0$ iff*

$$\forall \boldsymbol{x}. \ ||\boldsymbol{x} - \boldsymbol{x}_0||_\infty \leq \epsilon \Rightarrow \arg\max(N(\boldsymbol{x})) = \arg\max(N(\boldsymbol{x}_0))$$

Intuitively, Definition 1 states that for input vector $\boldsymbol{x}$, which is at a distance at most $\epsilon$ from a fixed input $\boldsymbol{x}_0$, the network assigns to $\boldsymbol{x}$ the same label that it assigns to $\boldsymbol{x}_0$ (for simplicity, we use here the $L_\infty$ norm, but other metrics could also be used). When a network is *not $\epsilon$-local-robust at point*

$\boldsymbol{x}_0$, there exists a point $\boldsymbol{x}$ that is at a distance of at most $\epsilon$ from $\boldsymbol{x}_0$, which is misclassified; this $\boldsymbol{x}$ is called an *adversarial input*. In this context, *local* refers to the fact that $\boldsymbol{x}_0$ is fixed. Larger values of $\epsilon$ imply a larger distance from $\boldsymbol{x}_0$, and hence a stronger robustness guarantee if the property holds. Intuitively, in a DNN for image classification that is $\epsilon$-local-robust, small perturbations to $\boldsymbol{x}_0$, i.e., perturbations so small that a human would fail to detect them, should not result in a change of predicted class.

**Distinct Adversarial Robustness.** Classification networks assign a value to each output label, which expresses the level of confidence in that label; and the label with the highest confidence score wins. We are interested in an adversarial input $\boldsymbol{x}$ only if it is *clearly* misclassified, i.e., if $\boldsymbol{x}$'s assigned label received a significantly higher confidence score than that of the label assigned to $\boldsymbol{x}_0$. For example, if $\arg\max(N(\boldsymbol{x}_0)) \neq \arg\max(N(\boldsymbol{x}))$ but $c(\boldsymbol{x}_0) = 0.41$ and $c(\boldsymbol{x}) = 0.42$, $\boldsymbol{x}$ is not distinctly an adversarial input; whereas a case where $c(\boldsymbol{x}) = 0.8$ is clearly much more relevant. We refer to adversarial inputs whose assigned confidence value is sufficiently high (greater than some $\delta$) as *distinct adversarial inputs*, and refine Definition 1 to only consider them, as follows:

**Definition 2.** *A DNN $N$ is $(\epsilon, \delta)$-locally-robust at input point $\boldsymbol{x}_0$, iff*

$$\forall \boldsymbol{x}. \ \|\boldsymbol{x} - \boldsymbol{x}_0\|_\infty \leq \epsilon \Rightarrow (\arg\max(N(\boldsymbol{x})) = \arg\max(N(\boldsymbol{x}_0)) \vee c(\boldsymbol{x}) < \delta)$$

## 3 THE PROPOSED METHOD

### 3.1 PROBABILISTIC ROBUSTNESS

Definitions 1 and 2, which are fairly common, are geared for a malicious adversary: they are concerned with the existence of an adversarial input, implicitly assuming the adversary will be successful in finding it if such an input exists. Here, we focus instead on a *random* setting, where perturbations can occur naturally, and are not necessarily malicious. We argue that this setting is more realistic for widely-deployed systems, such as medical devices, aerospace, and trains, which are expected to operate at a large scale for a prolonged period, and are more likely to randomly encounter adversarial inputs than those crafted by a malicious adversary. In this case, a natural method for assessing a model's robustness is to randomly perform input perturbations, and check whether these result in adversarial inputs, i.e., cause misclassification.

Towards this end, we propose to use a *probabilistic* measure of robustness, which signifies the probability of randomly perturbing $\boldsymbol{x}_0$ into an input which is *not* a distinct adversarial input:

**Definition 3.** *The $(\delta, \epsilon)$-probabilistic-local-robustness score of a DNN $N$ at input point $\boldsymbol{x}_0$, abbreviated $plr_{\delta,\epsilon}(N, \boldsymbol{x}_0)$, is defined as:*

$$plr_{\delta,\epsilon}(N, \boldsymbol{x}_0) \triangleq 1 - P_{\boldsymbol{x}:||\boldsymbol{x}-\boldsymbol{x}_0||_\infty \leq \epsilon} \left[(\arg\max(N(\boldsymbol{x})) = \arg\max(N(\boldsymbol{x}_0)) \vee c(\boldsymbol{x}) < \delta)\right]$$

The key point is that probabilistic robustness, as defined in Definition 3 is a scalar value; the closer this value is to 1, the less likely it is a random perturbation to $\boldsymbol{x}_0$ would produce an adversarial input. This is in contrast to Definitions 1 and 2, which are Boolean in nature. We also note that the probability value in Definition 3 can be computed with respect to values of $\boldsymbol{x}$ drawn according to any input distribution of interest. For simplicity, unless otherwise stated, we will assume that $\boldsymbol{x}$ is drawn uniformly at random.

To assess a DNN's probabilistic robustness using Definition 3, we need to measure how many inputs are in the $\epsilon$-ball around $\boldsymbol{x}_0$ are adversarial. Estimating this measure directly, e.g., with the Monte Carlo or Bernoulli Hammersley (2013) methods, is not feasible due to the typical extreme sparsity of adversarial inputs, and the large number of samples required to achieve reasonable accuracy Webb et al. (2018). Thus, we require a different statistical approach to obtain this measure, using only a small number of samples. We next propose such an approach.

### 3.2 THE NORMAL DISTRIBUTION

Our goal is to measure the probability of randomly encountering an adversarial input, by looking at a finite set of perturbed samples around $\boldsymbol{x}_0$. Each sample is created by perturbing the input features of $\boldsymbol{x}_0$, so that the overall perturbation size does not exceed $\epsilon$; and its adversariality is determined

according to Definition 2. The main question is how to extrapolate a conclusion regarding the whole population from these samples.

The normal distribution is a useful notion in this context: if we know that the perturbed inputs are distributed normally, it is straightforward to obtain such a measure, even if adversarial inputs are scarce.

To illustrate this point, we trained a VGG16 DNN model, and examined an arbitrary point $x_0$, classified as some label $l_0$, from its training set. We randomly generated 10,000 perturbed images from $x_0$, and ran them through the DNN. For each output vector obtained this way, we collected the highest confidence score assigned to any label other than $l_0$; these are plotted as the blue histogram in Figure 1. The green curve represents the normal distribution using the average and standard deviation of the raw data. As the figure shows, the data is normally distributed; and this claim is supported by running a "goodness-of-fit" test (explained later). Our goal is to compute the probability of a fresh, randomly-perturbed input to be misclassified, with a confidence score that exceeds a given $\delta$, say 0.6. For data distributed normally, as in this case, we begin with calculating the *statistical standard score* (*Z-Score*), which is the number of standard deviations by which the value of a raw score (in our case, $\delta$) exceeds the mean value. Once the Z-score is obtained, we can use the normal distribution function, which computes the correct probability of the event using the Gaussian function. In our case, we get $x_0 \sim \mathcal{N}(\boldsymbol{\mu} = 0.473, \boldsymbol{\Sigma} = 0.053^2)$, where $\boldsymbol{\mu}$ is the average score and $\boldsymbol{\Sigma}$ is the variance. The resulting Z-score is $\frac{\delta - \mu}{\sigma} = \frac{0.6 - 0.473}{0.053} = 2.396$, where $\sigma$ is the standard deviation. Finally, we get:

$$
\begin{aligned}
\text{plr}_{0.6, 0.04}(N, x_0) &= 1 - \text{NormalDistribution(Z-score)} \\
&= 1 - \text{NormalDistribution}(2.396) \\
&= 1 - \frac{1}{\sqrt{2\pi}} \int_{-\infty}^{t=2.369} e^{\frac{-t^2}{2}} dt \\
&= (1 - 0.008288) = 0.991712
\end{aligned}
$$

Thus, in this example, a perturbed image drawn uniformly at random has a chance of 0.82% of constituting an adversarial input.

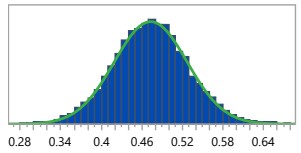

Figure 1: Normally-distributed adversarial inputs.

Of course, given data obtained empirically, as in our case, we need a way to determine whether the data is distributed normally before applying the aforementioned approach. A *goodness-of-fit* test is a procedure for determining whether a set of $n$ samples can be considered as drawn from a specified distribution. A common goodness-of-fit test for the normal distribution is the Anderson-Darlin Anderson (2011) test, which is used by widespread statistical, commercial applications such as SPSS IBM (2001) and JMP SAS (2001). Usually, a few hundred samples are more than enough to prove normal distribution with the Anderson-Darlin test.

## 3.3 THE BOX-COX TRANSFORMATION

Unfortunately, most often the adversarial inputs around $x_0$ are not normally distributed, and so the aforementioned approach does not immediately apply. For example, in our VGG16 model, out of the 10,000 points in the training set, only 1131 had normally-distributed adversarial inputs (as determined by the Anderson-Darlin test). Figure 2a illustrates the abnormal distribution for one of these input points, where we consequently cannot use the normal distribution function to estimate the probability of adversarial inputs in the population.

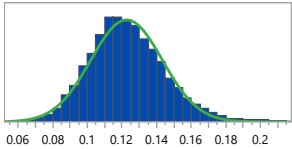 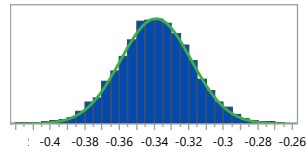

(a) Adversarial inputs that are *not* normally distributed.

(b) The now normal distribution, after applying Box-Cox.

Figure 2: Adversarial inputs that are initially not normally distributed.

The strategy that we propose for handling abnormal distributions of data like the one depicted in Figure 2a is to apply *statistical transformations*, capable of converting the given distribution into a normal one. This practice is widely used by statisticians, and is common in standard statistical applications such as SPSS and JMP. There are two main transformations used to normalize probability distributions: Box-Cox Box & Cox (1982) and Yeo-Johnson Yeo & Johnson (2000). Here, we focus on the Box-Cox power transformation, which is preferred for distributions of positive values (as in our case). Box-Cox is a continuous, piecewise-linear power transform function, parameterized by a real-valued $\lambda$, defined as follows:

**Definition 4.** *The Box-Cox$_\lambda$ power transformation of input $x$ is:*

$$Box\text{-}Cox_\lambda(x) = \begin{cases} \frac{x^\lambda - 1}{\lambda} & if\, \lambda \neq 0 \\ \ln(x) & if\, \lambda = 0 \end{cases}$$

Selecting the parameter $\lambda$ in often performed using the *maximum-likelihood estimation* (MLE) method. In this method, $\lambda$ is chosen by heuristically maximizing the goodness-of-fit score of the resulting distribution, so that it most closely resembles a normal distribution Rossi (2018); Asar et al. (2017).

Figure 2b depicts the distribution of the data from Figure 2a after applying the Box-Cox transformation, with an automatically calculated $\lambda = 0.257$ value. As the figure shows, the data is now normally distributed: x $\sim \mathcal{N}(\boldsymbol{\mu} = -0.33, \boldsymbol{\Sigma} = 0.020^2)$; this is confirmed by the Anderson-Darlin test with a confidence score of over 99%. Following the Cox-Box transformation, we can now calculate the Z-Score, which gives 7.45; and the corresponding plr score, which turns out to be extremely high: $1 - 4.66 \cdot 10^{-14}$.

### 3.4 THE ROMA CERTIFICATION ALGORITHM

Based on the previous sections, our algorithm for computing plr scores is given as Algorithm 1.

---

**Algorithm 1** Compute Probabilistic Local Robustness($\delta, \epsilon, n, N, \boldsymbol{x}_0, \mathcal{D}$)

---

1: **for** $i := 1$ to $n$ **do**
2:     $x_0^i$ = CREATEPETURBEDPOINT($x_0, \epsilon, \mathcal{D}$)
3:     confidence[i] $\leftarrow$ PREDICT($N, x_0^i$)
4: **if** ANDERSON-DARLIN(confidence) $\neq$ NORMAL **then**
5:     confidence $\leftarrow$ BOX-COX(confidence)
6:     **if** ANDERSON-DARLIN(confidence) $\neq$ NORMAL **then**
7:         **return** FAIL
8: avg $\leftarrow$ AVERAGE(confidence)
9: std $\leftarrow$ STDDEV(confidence)
10: z-score $\leftarrow$ Z-SCORE(avg,std,$\delta$)
11: **return** $1-$ NORMALDISTRIBUTION(z-score)

---

The inputs to the algorithm are: $\delta$, the confidence level of a distinct adversarial input; $\epsilon$, the maximum perturbation that can be added to $x_0$; $n$, the number of perturbed samples to generate around

$x_0$; $N$, the neural network, and $x_0$, the input point whose plr score is being computed; and $\mathcal{D}$, the distribution from which adversarial inputs are drawn. The algorithm starts by generating $n$ perturbed inputs of the provided $\boldsymbol{x}_0$, each drawn according to the provided distribution $\mathcal{D}$ and with a perturbation that does not exceed $\epsilon$ (lines 1–3). Next, on lines 4–7, it confirms that the samples' confidence values distribute normally, possibly applying the Box-Cox transformation if needed. Finally, on lines 8–11, the algorithm calculates the probability for a random adversarial input using the properties of the normal distribution, and returns the computed $\text{plr}_{\delta,\epsilon}(N, \boldsymbol{x}_0)$ score.

**Soundness and Completeness.** Algorithm 1 depends on the distribution of adversarial inputs being normal. If the distribution is initially not normal, the algorithm attempts to normalize it using the Box-Cox transformation. The Anderson-Darlin goodness-of-fit tests ensure that the algorithm will not treat an abnormal distribution as a normal one, and thus guarantee the soundness of the computed plr scores.

The algorithm's completeness depends on its ability to always obtain a normal distribution. As our evaluation (described later) demonstrates, the Box-Cox transformation can indeed accomplish this very often. However, the transformation might fail in producing a normal distribution; this will be identified by the Anderson-Darlin test, and our algorithm will stops with a failure notice in such cases. In that sense, Algorithm 1 is incomplete. In practice, failure notices by the algorithm can sometimes be circumvented — by increasing the sample size, or by evaluating the robustness of other input points.

In our evaluation, we observed that the success of Box-Cox often depends on the value of $\epsilon$. An analysis of the results indicated that very small or very large $\epsilon$ values more often led to failures, whereas mid-range values more often led to success. We speculate that this is because very small values lead to almost no adversarial inputs — i.e., the resulting distribution of adversarial inputs is close to uniform, and is consequently impossible to normalize. A similar situation occurs for very large $\epsilon$ values, which introduce a large number of adversarial inputs distributed uniformly. We argue that the remaining, mid-range values of $\epsilon$ are the more relevant ones. Adding better support for cases where Box-Cox fails, for example by using additional statistical transformations, remains a work in progress.

## 4 EVALUATION

For evaluation purposes, we implemented Algorithm 1 as a proof-of-concept tool. The tool is written in Python 3.7.10, and uses the TensorFlow 2.5 and Keras 2.4 frameworks. For our models, we used Resnet-10, Resnet-100, VGG16-10, VGG16-200 and Densenet, as described in Table 1, all trained using the CIFAR10 data set. All experiments mentioned in this section were run using the *Google Colab Pro* environment, with an NVIDIA-SMI 470.74 GPU and a single-core Intel(R) Xeon(R) CPU @ 2.20GHz. The code for the tool and experiments is (anonymously) available online Annonymized (2021), and will be publicly released with the final version of this paper.

Table 1: Neural network models' properties

| Model Name | Base Model | Reference | # Epochs | Accuracy | Loss |
|------------|------------|-----------|----------|----------|------|
| Resnet-10 | Resnet | He et al. (2016) | 10 | 0.72 | 1.07 |
| Resenet-100 | Resnet | He et al. (2016) | 100 | 0.91 | 0.4456 |
| VGG16-10 | VGG16 | Simonyan & Zisserman (2015) | 10 | 0.73 | 0.8329 |
| VGG16-200 | VGG16 | Simonyan & Zisserman (2015) | 200 | 0.76 | 2.7082 |
| Densenet | Densnet | Huang et al. (2017a) | 200 | 0.93 | 0.5335 |

### 4.1 EXPERIMENT 1: MEASURING ROBUSTNESS' SENSITIVITY TO PERTURBATION SIZE

By our notion of robustness given in Definition 3, it is likely that the $\text{plr}_{\delta,\epsilon}(N, \boldsymbol{x}_0)$ score decreases as $\epsilon$ increases. For our first experiment, we set out to measure the rate of this decrease. Using our Densenet model, we repeatedly invoked Algorithm 1 to compute plr scores for increasing values of $\epsilon$. For our $\boldsymbol{x}_0$, we arbitrarily selected the first 100 images from the CIFAR10 test set, and measured the average robustness of the images for each $\epsilon$. The averaged results (depicted in Figure 3) indicate

an almost linear correlation between $\epsilon$ and the robustness score. This result is supported by earlier findings Webb et al. (2018).

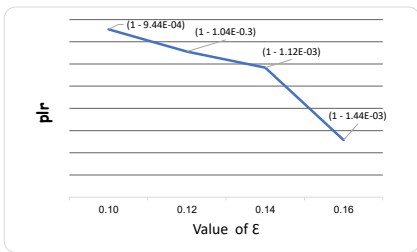

Figure 3: Average probabilistic robustness as a function of $\epsilon$.

The experiment was conducted by running Algorithm 1 with $\delta = 0.6$ on each of the 100 input images, and generating 10,000 perturbed samples for each image. Running the algorithm took less than 7 seconds per sample, and under one hour for the entire experiment. We note here that Algorithm 1 naturally lends itself to additional parallelization, as each perturbed input can be evaluated independently of the others; we leave adding these capabilities to our proof-of-concept implementation for future work.

## 4.2 EXPERIMENT 2: COMPARING ROBUSTNESS ACROSS MODELS

The ability to efficiently compute plr scores allows us to compare multiple models based on their robustness. Using Algorithm 1, we computed the plr scores for each of our 5 models, averaged over the first 100 images from the CIFAR10 test set. We arbitrarily set $\epsilon = 0.04$ and $\delta = 0.6$. The average plr scores appear in Figure 4, and indicate that a higher number of epochs leads to a higher robustness score.

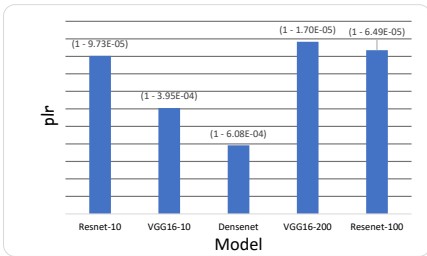

Figure 4: Robustness comparison between models

Running Algorithm 1 as part of this experiment with 10,000 perturbation samples for each image took between five seconds (with the VGG16-10 model) to seven seconds (with the Densenet model), for each sample. The whole experiment lasted less than one hour.

## 4.3 EXPERIMENT 3: CATEGORIAL ROBUSTNESS

For our third experiment, we focused on *categorial robustness*, where we first measure the robustness of inputs labeled as a specific category, and then compare the robustness scores across categories. We ran Algorithm 1 on our VGG16-10 model, for all 10,000 CIFAR10 test set images. We calculated the average robustness score for each of the ten categories separately. Table 2 depicts the results.

The results expose an interesting insight, namely the high variability in robustness between the different categories. For example, the average robustness score for inputs classified as Bird is more than ten times that of inputs classified as Dog. We applied a *T-test* and a *binomial test*, which are a well-established statistical tools for measuring the difference between two sets of values, to the Bird and Frog categories. The tests produced a similarity score of less than 0.3%, indicating that the

Table 2: Pivot analysis of all samples in the test set

| Category | # Samples | Robustness Avg | Robustness variance |
|----------|-----------|----------------|---------------------|
| Airplane | 1000 | $1 - 2.8894 \cdot 10^{-4}$ | $1.1362 \cdot 10^{-5}$ |
| Automotive | 1000 | $1 - 8.3363 \cdot 10^{-5}$ | $1.0430 \cdot 10^{-6}$ |
| Bird | 1000 | $1 - 1.1877 \cdot 10^{-4}$ | $1.5064 \cdot 10^{-6}$ |
| Cat | 1000 | $1 - 7.0537 \cdot 10^{-5}$ | $7.8409 \cdot 10^{-7}$ |
| Deer | 1000 | $1 - 1.3545 \cdot 10^{-4}$ | $1.9256 \cdot 10^{-6}$ |
| Dog | 1000 | $1 - 7.1440 \cdot 10^{-6}$ | $3.8810 \cdot 10^{-9}$ |
| Frog | 1000 | $1 - 6.6058 \cdot 10^{-5}$ | $6.0929 \cdot 10^{-7}$ |
| Horse | 1000 | $1 - 2.5776 \cdot 10^{-4}$ | $7.9067 \cdot 10^{-6}$ |
| Ship | 1000 | $1 - 1.6059 \cdot 10^{-4}$ | $3.0538 \cdot 10^{-6}$ |
| Truck | 1000 | $1 - 2.7746 \cdot 10^{-4}$ | $6.8932 \cdot 10^{-6}$ |

two categories are indeed distinctly different. From this, we draw the important conclusion that the per-category robustness of models can be far from uniform.

To the best of our knowledge, this is the first time such extreme differences in categorial robustness have been reported; and we believe this insight could affect DNN certification efforts, by allowing engineers to require separate robustness thresholds for different categories. For example, in a DNN for traffic signs recognition, a user might require a high robustness score for a stop sign and a low robustness score for a parking sign.

Running Algorithm 1 on the entire CIFAT10 test set with one thousand samples per input, using $\delta = 0.6$ and $\epsilon = 0.04$, took under 71 minutes.

**Completeness Level.** We observed that the completeness rate (the number of complete samples divided by the number of samples) varies between the models: in VGG16-10 and VGG-200, the completeness rate was 77% and 26% respectively; in Resnet-10 and Resnet-100 the rate was 70% and 57%, respectively; and in Densenet, the rate was 30%. In order to improve these rates, the steps described in Section 3.4 can be applied.

## 5 SUMMARY AND DISCUSSION

### 5.1 SUMMARY

In this paper, we introduced RoMA — a novel statistical and scalable method for measuring the probabilistic local robustness of a black-box DNN model. We demonstrated RoMA's applicability in several aspects and on multiple common DNN models. The key advantages of RoMA over existing methods are: (i) it uses straightforward and intuitive statistical method for measuring DNN robustness; (ii) it is scalable; (iii) it works on black-box DNN models, and makes no assumptions such as Lipschitz continuity or piecewise-linear constraints; and (iv) the method is quick in comparison to formal verification methods and other methods that require hours or more for analyzing local robustness Wang et al. (2018). The key limitation of our approach is that it depends on the normal distribution of the adversarial inputs, and will fail whenever the Box-Cox transformation does not normalize this distribution.

The plr scores computed by RoMA indicate the risk of using a DNN model, and can allow regulatory agencies to conduct *risk mitigation* procedures: a common practice for integrating sub-systems with a known MTBF into safety-critical systems. The ability to perform risk and robustness assessment is an important step towards using DNN models in the world of safety-critical applications, such as medical devices, UAVs, automotive, and others. We believe that the newly defined notion of *categorial robustness* could also play a key role in this endeavor.

Moving forward, we intend to: (i) evaluate our tool on additional norms, beyond $L_\infty$; and (ii) better characterize the cases where the Box-Cox transformation fails, and search for other statistical tools can succeed in those cases; and (iii) improve the scalability of our tool by adding parallelization capabilities.

**Ethics Statement** This paper does not raise any ethical concerns.

**Reproducibility Statement** All the details required to reproduce the results for this paper are detailed in the evaluation part of this paper in section 4. The code for the tool and experiments is (anonymously) available online Annonymized (2021), and will be publicly released with the final version of this paper.

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
