# OpenReview forum: "RoMA: a Method for Neural Network Robustness Measurement and Assessment "
_ICLR.cc/2022/Conference — ICLR 2022 Submitted_

### Official Review · Reviewer_Y2p2 · 2021-11-01

**Correctness:** 2
**Technical Novelty And Significance:** 2
**Empirical Novelty And Significance:** Not applicable
**Recommendation:** 3
**Confidence:** 4

**Main Review:**

Pros:


1. The paper tries to address an important problem in the field of adversarial machine learning: how to evaluate
robustness of deep neural networks efficiently, which is of great importance and has wide application.


2. Experiments show that the proposed statistic decreases as epsilon (the distortion bound) increases, which shows
that the proposed statistic behaves expectedly. Since larger epsilon usually leads to weaker robustness (as the
probability of adversarial examples existing improves), a reasonable statistic that measures robustness should
decrease as epsilon increases.


3. It's nice to see a comparison of robustness of different architectures.
Code is provided for reproducing the experimental results.


##########################################################################

Cons:

1. Definition 3 is problematic. Based on the sentence before the definition "..., which signifies the probability of
randomly perturbing $x_0$ into an input which is not a distinct adversarial input", the robustness score should
be $1-$probability of perturbed input being an adversarial input. Thus, the definition of score should be:
\begin{align*}
plr_{\delta, \epsilon}(N,x_0):=1-P_{x:\|x-x_0\|\le\epsilon}[argmax(N(x))\ne argmax(N(x_0)) \bigcap c(x)>\delta].
\end{align*}
The definition in the paper does not satisfy the claim "the closer this value is to 1, the less likely it is a random
perturbation to $x_0$ would produce an adversarial input." Based on the original definition, the closer the value
is to 1, the closer the value of $P_{x:\|x-x_0\|\le\epsilon}[argmax(N(x))=argmax(N(x_0)) \bigcup c(x)<\delta]$ is to 0,
which makes the probability of robustness close to 0.


2. The proposed method section is not clear. Based on the description of z-score calculation, my understanding
is that the method is trying to measure $1-P(c(x)>\delta)$, which is $1-$the probability of the maximum confidence
score of perturbed input greater than a preset threshold. Here're some problems of it:

- In this section, it says that $x_0\sim N(\mu=0.473, \Sigma=0.053^2)$. Isn't $x_0$ the original input? Why suddenly
it follows a normal distribution? Based on the context, I guess it represents the distribution of maximum confidence
scores of perturbed inputs. Another possibility is that it represents the distribution of maximum confidence scores of
adversarial examples out of the perturbed inputs. If my understanding is wrong, please correct me.

- If it represents the distribution of maximum confidence scores of perturbed inputs, why $1-P(c(x)>\delta)$ can be used to measure robustness? $P(c(x)>\delta)$ does not represent the probability of $x$ being adversarial. The probability of adversarial should be: $P[argmax(N(x))\ne argmax(N(x_0))\bigcap c(x)>\delta]$. Where is the requirement of $argmax(N(x))\ne argmax(N(x_0))$ reflected in the method?

- Even if the authors compute the empirical distributions of maximum confidence scores based on only adversarial
examples out of the perturbed samples, it's only a conditional probability and the component of $P[argmax(N(x))\ne argmax(N(x_0))]$
is still missing. Besides, estimating $P[argmax(N(x))\ne argmax(N(x_0))]$ is a difficult problem. The probability cannot simply be estimated by number of adversarial examples in the perturbed inputs divided by total number of perturbed inputs, because we don't know how adversarial samples are distributed in the input space.

3. For experiment 4.1, why not report plr scores of $\epsilon<0.1$? On CIFAR10, a white-box attack like PGD with perturbation budget $0.03$ can reduce the accuracy to $0$. So, it would be better to report the plr scores for $\epsilon<0.1$ as well.

4. The paper uses "adversarial inputs" many times. In the abstract, "adversarial inputs" is defined as: "small input perturbations that cause the model to produce erroneous outputs." However, later on, it is used to describe maximum confidence scores of adversarial examples. For example, in Figure 1, the caption is "Normally-distributed adversarial inputs". Based on the description, the histogram is generated with highest confidence scores assigned to any label other than original label. So, a better caption can be "Histogram of highest confidence scores of adversarial examples of 10,000 perturbed images". This happens several times, making the definition of adversarial inputs very confusing.

##########################################################################

Minor:

1. "CONFERENCE SUBMISSION" should be removed from the title.

2. Page 5, when calculating $plr_{0.6, 0.04}(N,x_0)$, the value $0.008288$ implies that upper tail value is calculated, but the integration represents lower tail.

**Summary Of The Paper:**

Summary:

The paper introduces a statistical method to measure the robustness of deep neural networks. The novel
part of the method is that it's designed to measure the probability of random points near an input being
adversarial, instead of probability of adversarial examples existing in the vicinity of an input. To measure it,
the authors proposes to use Box-cox transformation to transform distribution of confidence scores to normal,
then calculate the probability based on it.



**Summary Of The Review:**

Reasons for score:


Overall, I vote for rejection. I like the idea of measuring the probability of random inputs being adversarial
instead of finding the extreme case. However, I think whether the proposed method is measuring such
probability is questionable. Some notations and terms are unclear, so it is possible that I missed some
components. If the authors can address my major concerns in the rebuttal period, I'm ok to increase
my score and accept the paper.

---

### Official Review · Reviewer_uGd4 · 2021-11-01

**Correctness:** 2
**Technical Novelty And Significance:** 1
**Empirical Novelty And Significance:** 1
**Recommendation:** 3
**Confidence:** 5

**Main Review:**

This paper has several drawbacks, which are detailed as follows.

1. There are several existing methods that can provide similar probabilistic robustness measures. Although the authors emphasize that the proposed method uses random sampling to evaluate robustness and obtain a statistical robustness score. Similar ideas are already widely used in the literature. For example, randomized smoothing [R1], which is one of the most widely used certification techniques, already provides probabilistic bounds and confidence intervals for local robustness assessment. Moreover, [Weng 2019] considered the problem setting of a random perturbation to a local data sample. The resulting bound is also similar to the proposed local sampling technique. Since the idea is similar and the authors did not compare to those existing methods, I don't see many insights from the proposed method compared to existing works.

2. The experimental results are not convincing. First, the authors did not provide any empirical robustness results to compare with the robustness evaluations. How can one know the reported results are correct and meaningful? Without rigorous justification and empirical evidence, the results could be questionable. Second, the authors did not compare to existing methods, which makes it difficult to assess the quality of the proposed method.


[R1] https://arxiv.org/abs/1902.02918



**Summary Of The Paper:**

This paper proposes RoMA, a robustness evaluation framework based on local sampling and probability computation.

The main contributions are:
1. Proposal of the ($\epsilon$,$\delta$) local robustness score for assessing the probability of random local samples that have different predictions than a given data input with a $\delta$-confined top-1 confidence.

2. Use of Box-Cox transformation for input data to improve statistical estimation.

3. The method can be implemented in a model-agnostic fashion.


**Summary Of The Review:**

The current version is quite incomplete. It lacks comparisons and distinctions to prior arts for technical novelty, and it also lacks comparisons to empirical robustness and other baseline methods.

---

### Official Review · Reviewer_FezE · 2021-11-02

**Correctness:** 3
**Technical Novelty And Significance:** 2
**Empirical Novelty And Significance:** 2
**Recommendation:** 5
**Confidence:** 4

**Main Review:**

Strengths:

+ The paper addresses the very important challenge of characterizing robustness of deep learning models.

+ The described approach is simple - transform the neighborhood confidence c_i values to a normal distribution  and then estimate the z-score.

Weaknesses:

- Some observations are rather obvious and well-known. Different classes exhibit different robustness. Confusion matrices have been used widely to understand these class-conditional error distributions. The reviewer is not convinced that "categorical robustness" is a new notion.

- The decision of normalizing c^i on sampled input appears arbitrary. How do we decide what inputs to sample? The model is likely going to be robust in some part of the input space and not robust in another. Sampling inputs on which the perturbation is applied appears a crucial decision. Evaluating the model in just the neighborhood of the training data is not very useful.

- Could authors provide more details on how lambda was selected? The paper states that "Selecting the parameter λ in often performed using the maximum-likelihood estimation (MLE) method. In this method, λ is chosen by heuristically maximizing the goodness-of-fit score of the resulting distribution, so that it most closely resembles a normal distribution." This appears to describe the usual practice. In this paper how was λ selected? Was a specific tool used to implement this?

- "An analysis of the results indicated that very small or very large  values more often led to failures, whereas mid-range values more often led to success. We speculate that this is because very small values lead to almost no adversarial inputs — i.e., the resulting distribution of adversarial inputs is close to uniform, and is consequently impossible to normalize. A similar situation occurs for very large  values, which introduce a large number of adversarial inputs distributed uniformly". Could authors clarify what is meant by uniformly distributed? It is not intuitively obvious why adversarial examples would be uniformly distributed in the input space. The decision of the model would be more sensitive with respect to some features (or pixels) than others. Also, given the observations that the technique only works on some mid-range values of epsilon. Could authors comment on how this limitation would impact its practical utility?

- A minor suggestion - forward references such as "explained later" should be avoided. Putting a citation for standard textbook approaches such as goodness-of-fit tests would suffice.

**Summary Of The Paper:**

The paper presents a statistical method - Robustness Measurement and Assessment (RoMA) to measure the expected robustness of a neural network model. The robustness is defined as the probability that a random input perturbation causes an incorrect prediction. The presented approach is a blackbox approach. Different output labels are observed to exhibit different robustness values.

The basic premise of the paper is that the adversarial perturbations are not naturally normal, but a transformation (Box-Cox) can be applied to make them a normal distribution before applying statistical estimation techniques (Anderson-Darling test + z score).

**Summary Of The Review:**

The paper presents a simple approach to characterize robustness. Some limitations of the techniques need to be clarified for the reviewer to understand the contribution and limitations of the proposed approach.

---

### Official Review · Reviewer_JNVz · 2021-11-02

**Correctness:** 3
**Technical Novelty And Significance:** 3
**Empirical Novelty And Significance:** 3
**Recommendation:** 3
**Confidence:** 2

**Main Review:**

While the paper appears quite novel and interesting, I did not feel the text reflected the abstract as a whole. The paper was quite opaque, and I remained unconvinced that the method if applied in practice would provide useful insights.

- Needs proofreading, the repeated citations in the related work section is distracting.
- Additional motivation for Definition 3 is warranted. Can the authors present a practical motivating example, which does, or could believably, occur in reality?
- Grammatical errors: "we need to measure how many inputs are in the $\epsilon$-ball around $x_0$ are adversarial"
- "To illustrate this point, we trained a VGG16 DNN model," on what dataset? with what training parameters? These details should at least be put in the appendix. How exactly are the perturbations generated, what attack, what attack parameters?
- While I appreciate the acknowledgement that most often the adversarial inputs around a datapoint are not normally distributed, the example in Figure (2a) is very close to normal. I would hypothesise that a wide variety of distributions are possible given different parameterisations of  dataset, downstream task, label distribution, architecture, training parameters altogether. On ImageNet, if the perturbations are generated by non-targeted attacks, it is significantly easier to induce misclassification via related categories (different types of dogs) than classes like "frogs". Altogether, I am unconvinced that the figures as presented correspond to expected scenarios in reality, and even if so, I expect the variance across datasets and training parameters to be high.
- Does the Box-Cox transformation preserve the statistics of the distribution, what statistics are and aren't preserved? From a functional perspective, can the transformation be considered bijective, can one reverse the Box-Cox transformation and recover the original distribution. One can transform one distribution to an entirely different distribution, but that isn't interesting if the transformed distribution does not reflect the original distribution in any unique way? This should be clarified in the text.

**Summary Of The Paper:**

Present a method to measure the expected robustness of a neural network model, by determining  the probability that a random input perturbation might cause misclassification, providing formal guarantees regarding the expected frequency of errors that a trained model will encounter after deployment. The method can be applied black-box. Applied the approach to compare the robustness of different models, and measure how a model’s robustness is affected by the magnitude of input perturbation.

**Summary Of The Review:**

Recommend rejection as the paper did not demonstrate its utility in practice, nor corroborated their abstract in a clear manner.

---

### Decision · Program_Chairs · 2022-01-20

**Decision:**

Reject

**Comment:**

This paper introduces a technique to measure the *expected* robustness of a
neural network by measuring the probability random input perturbations will
cause the model to make a mistake.

The reviewers are not convinced by the results in this paper. The methods
are not carefully evaluated against prior work, and it is not exactly
clear what lesson one can draw from the resulting statistical evaluation.
The experimental setup is not clearly explained in several places, making the
paper difficult to fully follow.

Since the authors do not respond to the reviewer concerns, there was no
opportunity to address these concerns.